# Advances in Chitosan-Based CRISPR/Cas9 Delivery Systems

**DOI:** 10.3390/pharmaceutics14091840

**Published:** 2022-09-01

**Authors:** Anna E. Caprifico, Peter J. S. Foot, Elena Polycarpou, Gianpiero Calabrese

**Affiliations:** School of Life Sciences, Pharmacy and Chemistry, Kingston University London, Penrhyn Road, Kingston upon Thames, London KT1 2EE, UK

**Keywords:** chitosan, chitosan functionalisation, gene delivery system, CRISPR/Cas9, cancer, genetic disorders

## Abstract

Clustered regularly interspaced short palindromic repeat (CRISPR) and the associated Cas endonuclease (Cas9) is a cutting-edge genome-editing technology that specifically targets DNA sequences by using short RNA molecules, helping the endonuclease Cas9 in the repairing of genes responsible for genetic diseases. However, the main issue regarding the application of this technique is the development of an efficient CRISPR/Cas9 delivery system. The consensus relies on the use of non-viral delivery systems represented by nanoparticles (NPs). Chitosan is a safe biopolymer widely used in the generation of NPs for several biomedical applications, especially gene delivery. Indeed, it shows several advantages in the context of gene delivery systems, for instance, the presence of positively charged amino groups on its backbone can establish electrostatic interactions with the negatively charged nucleic acid forming stable nanocomplexes. However, its main limitations include poor solubility in physiological pH and limited buffering ability, which can be overcome by functionalising its chemical structure. This review offers a critical analysis of the different approaches for the generation of chitosan-based CRISPR/Cas9 delivery systems and suggestions for future developments.

## 1. Introduction

The development of genome-editing technologies for the treatment of genetic and acquired diseases, such as cancer, is a rapidly growing field [1]. Genome editing can insert, replace, or disrupt a DNA sequence at specific loci of the genome, in turn modifying gene products, so as to manage several disorders, including many types of cancer and other diseases with a genetic origin, such as cystic fibrosis [2]. Especially within genome-editing technologies, those based on programmable nucleases can introduce new opportunities for therapeutic genome editing due to their ability to remove or correct deleterious mutations or insert protective mutations [3], offering a powerful mean for biological research and treatment of several diseases such as cancer. Programmable nucleases insert double-strand break (DSB) at specific sites of the genome, following their repair by homology directed repair (HDR) or non-homologous end joining (NHEJ), according to the cell condition and the aid of a repair template: HDR employs an exogenous DNA template to repair the lesion, whereas NHEJ does not require a template since the lesion is repaired by connecting the two DSB ends [3]. Several programmable nucleases have been developed, including Zn-finger nucleases and meganucleases, which have as a drawback the fact that they recognise specific DNA sequences using a protein; hence, the engineering of new proteins is needed for each targeted gene [2]. Conversely, the clustered regularly interspaced short palindromic repeat (CRISPR) and the associated Cas endonuclease (Cas9) is a novel platform of programmable nucleases that requires a minimal adjustment for the target design, relying only on the engineering of single guide RNA (sgRNA) [3].

Using this novel approach, genome can be edited using three strategies: a plasmid based CRISPR/Cas9 system encoding the Cas9 protein and sgRNA, a mixture of the Cas9 messenger (m)RNA and the sgRNA, or a mixture of Cas9 protein and the sgRNA. The most simple and convenient approach is the plasmid based CRISPR/Cas9 system since the transfection of numerous components in the same cells is avoided. Moreover, it shows higher stability than mRNAs and is digested using a restriction enzyme, following ligation with an annealed oligonucleotide designed for a specific targeting site. However, this method shows several challenges: the direct delivery to the nucleus where it can be translated into Cas9 mRNA needs to be achieved. Moreover, plasmid-based CRISPR/Cas9 systems produce more off-target effects including small and large insertions comparing to the delivery of mRNAs [4]. Therefore, the most investigated strategy in the recent years is the delivery of the Cas9 protein complexed with sgRNA, in a complex known as ribonucleoprotein (RNP). The Cas9 protein is invariable. while being easily re-targeted by modifying only a small portion of the sequence belonging to the sgRNA. The sgRNA is made of two parts: crispr RNA (crRNA, which is complementary to the targeted DNA) and a tracr RNA (which acts as scaffolding for the Cas9 nuclease) (Figure 1). Once the target sequence is recognised by the sgRNA, the Cas9 introduces DSB on these site-specific genes, inducing breaks and damage to inactivate the target; DSB is then repaired by HDR or NHEJ [3]. This novel gene editing technology won the Nobel Prize in Chemistry in 2020 and represents the most efficient and suitable tool for gene editing.

Even though the CRISPR/Cas9 system is characterised by a simple design along with high versatility and flexibility, its targeted delivery is still challenging [5]. Indeed, it must overcome many barriers including cellular uptake of the complex, its escape from lysosomes degradation, and finally its nuclear translocation and translation of proteins. Moreover, the in vivo application of CRISPR/Cas9 is challenged by its large size (163 kDa) and the need of a safe and efficient delivery system able to carry and deliver the cargo to the nucleus for the regulation of the nuclease activity [6].

The CRISPR/Cas9 complex is currently delivered to cells by physical or viral methods [2]. Physical methods include electroporation or microinjection, which have as a drawback the limited in vitro or ex vivo use [2]. The use of viral vectors, including lentivirus or adenovirus, consists of integrating the CRISPR/Cas9 complex into the viral genome, which is then used to infect the cells. The viral vectors confer high delivery efficiency, but concerns arise about the safety of these types of carriers since they can lead to carcinogenesis, immunogenicity, or insertional mutagenesis [2,7]. Finally, viral vectors imply high costs, low loading capacity, specialised infrastructures, and a high level of expertise for their large-scale production [8,9].

Therefore, much research is focused on the optimisation of CRISPR/Cas9 delivery systems [10]. The consensus relies on the use of non-viral systems, which still show some disadvantages such as the likelihood of inducing off-target effect [10]. Hence, delivery systems able to control the CRISPR/Cas9 activity have been designed based on their ability to respond to specific stimuli such as light, pH, temperature, or magnetic field [5]. For instance, a specific light irradiation, which can vary from ultraviolet (UV) to near infrared (NIR) can be absorbed by chromophores, thus controlling the activity of Cas9. However, wavelengths shorter than 200 nm are rarely employed due to their cytotoxic effects and the photosensitivity of some molecules [7]. Moreover, UV and visible light are not able to penetrate deeply into the tissue and can be lethal at high doses. Conversely, NIR-triggered delivery systems do not have these disadvantages, and they offer enhanced specificity, reversibility, and quick response [5].

Recently, nanocarriers (NCs) such as nanoparticles (NPs) have shown exclusive advantages in gene delivery, especially for their targeted ability [11]. Indeed, the surface of NPs can be engineered to bind specifically to the targeted cell or tissue. In addition, NPs can protect the cargo from degradation in the systemic blood circulation and can deliver high molecular weight (MW) proteins, such as the 163 kDa CRISPR/Cas9 complex, directly into the nucleus of cells; finally, the materials used to manufacture NPs are safe and do not elicit immunogenicity [10]. Various CRISPR/Cas9 loaded NCs have been employed, including cationic liposomes, lipid NPs, gold NPs, exosomes, and NCs based on cationic polymers such as polyethyleneimine (PEI) and chitosan (Cs) [11]. The latter has attracted much attention for the delivery of nucleic acids since it derives from a natural polymer (chitin), hence it shows biodegradability, biocompatibility, and nontoxicity properties [12]. Several reviews have focused on the use of this polymer as the backbone of gene delivery systems [13,14]. However, owing to the novelty of the topic, the literature lacks a review that focuses on chitosan-based CRISPR/Cas9 delivery systems. Thus, this work presents an overview of the most recent advances made on the generation of efficient chitosan-based CRISPR/Cas9 delivery systems. The aim is to achieve a carrier characterised by high transfection efficiency, overcoming some chitosan limitations including buffering ability, poor specificity, and stability following intravenous administration. However, the transfection efficiency of most of the systems discussed has been tested in vitro, and further studies are needed. Currently, only few systems have been tested in mice, suggesting their suitability for further clinical investigations.

## 2. Chitosan in Gene Drug Delivery

Chitosan is made of repeated units of D-glucosamines and N-acetyl-D-glucosamines, which are linked by glycosidic bonds (Figure 2).

The D-glucosamines units possess primary amino groups that, below a given pK_a_ (6.5), become protonated and by doing so, confer to CsNPs multiple properties for gene delivery, which are illustrated in Figure 3.

Indeed, the positively charged chitosan can bind to negatively charged nucleic acids by electrostatic interactions so to form stable polyplexes [12]. Furthermore, since the cellular and nuclear membranes are negatively charged, the interaction with the positively charged chitosan leads to the uptake and nuclear relocation of the cargo. This process was proven by Ma et al. [15] who generated CsNPs, with a surface potential of about +20 mV, and employed them for the delivery of NF-kB/p65 antisense oligonucleotide to the nucleus of RAW264.7 macrophages.

The positively charged amino groups of chitosan are also involved in the endosome escape of the cargo since they can trigger a “proton sponge effect” (Figure 4). Indeed, following cellular uptake, a key step for gene carriers is to escape the endosomes that will integrate to lysosomes, ultimately leading to the destruction of the cargo due to the acidic pH and hydrolytic enzymes [16]. Conversely, cationic polymers such as chitosan and PEI can “consume” protons, becoming protonated and inducing the proton pumps to exhaustion. Indeed, following uptake of positively charged CsNPs, protons are transported into the endosomes, along with chloride and water, causing swelling and destruction of the endosome [13]. However, the buffering capacity of chitosan can be improved, in turn increasing the transfection efficiency. For instance, a carrier made of chitosan and RNA was coated with alginate to make the system more stable at physiological pH, so that the amino groups were protected and available for protonation in endosomes [17].

The positive charges on chitosan also confer the property of mucoadhesion, which allows CsNPs to be employed as a drug delivery system where the mucosal barrier needs to be overcome [18]. Indeed, the protonated amino groups of chitosan can interact with the negatively charged mucin to enhance the residence time at the target tissue, in turn increasing its absorption [19].

These properties make chitosan widely used in many biomedical applications, including gene delivery systems [20]. Nevertheless, its use is limited by the poor transfection efficiency, which is influenced by many factors such as the low solubility at physiological conditions, the degree of deacetylation (DDA), the MW, and the charge ratio of chitosan to the genetic material [13].

The functional groups of chitosan (primary amines as well as primary and secondary hydroxyl groups) allow chemical modifications to improve its properties. For instance, some strategies can help to improve its solubility, which is otherwise only achievable at pH values below 6.5. These strategies include the use of low MW derivatives of chitosan, which are often used as gene delivery vectors, even though some limitations arise, including poor specificity and very strong interaction with nucleic acid, resulting in limited unpackaging [21]. Other strategies to improve chitosan solubility include chemical modifications of its structure, such as the addition of trimethyl groups (*N*, *N*, *N*-trimethyl chitosan, and TMC), carboxymethyl groups or poly (ethylene glycol) (PEG and PEGylation). Chitosan’s structure can also be modified by the addition of target ligands to improve the specificity of the gene delivery system. These ligands include proteins, aptamers, carbohydrates, peptides, or other small molecules that enhance tumour targeting with low side-effects [22]. All these chitosan modifications will be discussed in the context of chitosan-based CRISPR/Cas9 delivery systems in the next section.

## 3. Chitosan-Based CRISPR/Cas9 Delivery Systems

### 3.1. Strategies Using Pristine Chitosan Backbone

Some recent CRISPR/Cas9 delivery systems based on chitosan were developed using its pristine chemical structure. For instance, CsNPs were employed by Nugrahaningsih et al. [23] to load the CRISPR/Cas9 complex for the treatment of pulmonary arterial hypertension (PAH), a disease associated mainly with genetic variations [24]. The main gene involved was suggested to be the bone morphogenic protein receptor II (BMPR2), which shows functional changes or reduced expression, leading to reduced proliferation and increased apoptosis of pulmonary cells. Therefore, the transfection of CRISPR/Cas9 for editing of BMPR2 was performed in fibroblasts. Results showed that the mRNA expression of BMPR2 was significantly decreased, resulting in increased cell proliferation [23], suggesting successful transfection of cells using pristine CsNPs (Figure 5A).

In another study by Srivastav et al. [25], the pristine structure of chitosan was used to coat NPs based on poly-(lactic-co-glycolic acid) (PLGA). This is a widely used polymer in the context of drug delivery systems. Due to its biocompatibility and biodegradability, it has been approved by the FDA for the delivery of therapeutics, but the absence of functional groups on its surface limits the active and passive targeting of the NCs. Coating the surface of PLGA NPs with chitosan would increase the electrostatic interactions with the negatively charged cell membranes allowing site-specific and controlled delivery [26]. Therefore, this system was used for the delivery of fluorescein isothiocyanate (FITC)-labelled CRISPR/Cas9 complex and, following cellular uptake studies, the complex appeared to be located inside the nucleus of cells. Moreover, the treatment with the CRISPR/Cas9 plasmid delivery system resulted in the 80% suppression of the expression of green fluorescence protein (GFP) in the human embryonic kidney cell line (HEK-293) [25] (Figure 5B).

Alallam et al. [27] employed the pristine structure of chitosan to coat the plasmid loaded alginate (AG) NPs. As a result of the electrostatic coating, the size of the NPs doubled from ~200 nm (AGNPs) to ~400 nm, and the zeta potential of NPs increased significantly from –4 mV (AGNPs) to about +30 mV. More importantly, the coating of AGNPs with Cs conferred a higher protection of the plasmid in the presence of serum proteins, although the formation of protein corona was induced since the size of NCs increased by 1.3-folds. Due to the protein corona formation, the cellular uptake decreased but the presence of chitosan on the surface of NPs enhanced the transfection ability by ~20% in several cancer cell lines, due to the protection of the plasmid from enzymatic degradation [27] (Figure 5C).

However, most of the research performed to generate chitosan-based CRISPR/Cas9 delivery systems focuses on boosting chitosan’s properties to improve the transfection efficiency. Therefore, efforts were made to obtain a vector showing improved specificity, high stability, boosted buffering ability, or a combination of more than one of these factors.

### 3.2. Strategies Improving the Specificity

#### 3.2.1. Magnetic Field

To increase specificity, Lee et al. [28] achieved the transcription of specific genes, such as cellular proto-oncogene (c-Myc), for cellular reprogramming in HEK-293 cells by binding transcription activators such as the VP64-p65Rta to the CRISPR complex, which was in turn immobilised on magnetic peptide-imprinted CsNPs (Figure 6A). Indeed, magnetic field-responsive NPs were shown to control drug release following the application of an external magnetic field, that, within the range of 350 to 400 kHz, is characterised by deep penetration into the tissues and poor absorption [5].

#### 3.2.2. Use of Ligands

In the context of hepatocellular carcinoma (HCC) treatment, Zhang et al. [29] developed NPs based on Cs conjugated to β-galactose carrying lactobionic acid (La), which specifically targets HCC cells expressing the asialoglycoprotein receptor (ASGPR) on their cellular membrane (Figure 6B) [30]. Paclitaxel, a commonly employed anticancer drug was loaded into the NPs in conjunction with the single-guided vascular endothelial growth factor receptor 2 (sgVEGFR2)/Cas9 plasmid, to potentiate paclitaxel’s effect [31]. In vitro studies showed that the presence of La on the vector boosted the ability of NPs to preferentially accumulate within ASGPR-overexpressing cells. Moreover, the loaded NPs were able to downregulate the expression of VEGFR2 and NF-kB p65 in liver cancer cells, HepG2. In vivo studies performed on hepatoma mice model resulted in more than 70% inhibition of tumour growth following treatment with this novel system. The genome editing efficiency was up to 39% in vitro and 33% in vivo. Moreover, systemic distribution and biosafety evaluation of the novel chitosan-based NPs was assessed by labelling them with the fluorescent compound, Cy5.5. In the tumour site, the fluorescence signal gradually increased up to 24 h, suggesting prolonged blood circulation of the carrier and no significant side effects on normal organs and tissues such as heart, liver, or spleen [29].

In a recent study performed by Khademi et al. [32], the targeted delivery efficiency of Cs-based CRISPR/Cas9-loaded NCs was induced by integrating two ligands (hyaluronic acid (Ha) and AS1411 aptamer on the surface of NPs (Figure 6C). Ha is a hydrophilic natural polymer targeting cluster of differentiation 44 (CD44) receptors expressed mainly on tumours of epithelial origin [33]. This targeting ability was boosted by the addition of the AS1411 aptamer, a well-known nucleolin (overexpressed in cancer cells) targeting group [34]. The CRISPR/Cas9 system employed was for FOXM1 knockout, being an oncogenic transcription factor upregulated in several cancers [35]. Gene transfection in vitro studies were performed on several cancer cells including MCF-7, HeLa and SK-MES-1 cells, and a human embryonic kidney cell line, HEK-293, used as a control. Results showed that the transfection efficiency ranged between 20 and 30% in cancer cells, while being 4% in HEK293 cells. Therefore, the cell viability was lower in cancer cells, ranging between 37 and 56%, while being about 88% in HEK-293 cells. Finally, in vivo studies on tumour-bearing mice suggested suppression of tumour growth (up to 90%) and increased survival time, following treatment with the novel delivery system compared to the control (e.g., naked plasmid or buffer saline). Altogether these results suggested a boosted targeting ability toward the tumour, with decreased off-target effects of the novel CRISPR/Cas9 delivery system [32].

### 3.3. Strategies Increasing the Stability

#### 3.3.1. Chitosan Tetrazole

Rabiee et al. [36] employed chitosan tetrazole (CsTz) (Figure 7A), which possesses a higher proportion of amines and imines than pristine chitosan, to generate a non-viral gene delivery vector carrying CRISPR/Cas9. The tetrazole functional groups on chitosan were involved in the interaction with the negatively charged phosphate backbone of CRISPR, especially at physiological pH. Polyplexes were formed by a self-assembly method at different weight ratios of CsTz/CRISPR (8:1 and 2:1). Those nanocomplexes showed a strongly positive surface potential (ranging between +45 and +39 mV) which implied high stability, and efficient cellular uptake along with low cell viability. In vitro gene expression efficiency was carried out in the HEK-293 cell line by investigating the expression of the enhanced GFP, which increased in proportion to the weight ratio CsTz/CRISPR, reaching values higher than 25%. These results suggested that CsTz based vectors may represent low cost and highly efficient CRISPR/Cas9 delivery carriers, although the strongly positive charges may induce the formation of protein corona in vivo [37].

#### 3.3.2. PEGylation

As mentioned, PEGylation is a widely used method to increase chitosan solubility in a broad range of pH conditions. PEGylation of NCs is also widely employed to minimise opsonisation and their clearance by the reticuloendothelial system (RES), in turn enhancing tumour targeting [37]. Since PEGylation enhances the stability of CsNPs, this chemical modification of chitosan is widely used to increase gene transfection of CsNPs [38,39]. However, PEGylation leads to a “PEG dilemma” since the coating of the gene/drug carrier with PEG limits its cellular interaction and internalisation, affecting the cargo’s delivery [40]. Furthermore, anti-PEG antibodies can be generated, leading to clearance of the gene carrier, in turn reducing its therapeutic efficiency [41]. Several derivatives of PEG have been developed in the coating of CsNPs to overcome these limitations, including carboxymethylated PEG (100) monostearate or methoxy-PEG [37].

PEG coating was also shown to enhance the diffusion rate of NCs through various types of mucus, such as cystic fibrosis mucus [42]. With the aim of increasing the transfection efficiency of CRISPR/Cas9 loaded CsNPs through the thick pulmonary mucus, Zhang et al. [12] conjugated a derivative of PEG, monomethyl ether (mPEG), to the hydroxyl group of chitosan (Figure 7B). In this way, the pulmonary gene delivery was achieved because the positive amine groups involved in the condensation of DNA were preserved, while boosting chitosan’s mucoadhesive property [12]. The mucus permeation ability of the system was evaluated in vitro through a transwell assay, and the transfection efficiency was tested in the HEK-293 embryonic cell line. Results showed that the coating with mPEG allowed an improved penetration of the nanocomplex compared to CsNPs with no mPEG coating. Moreover, the highest transfection efficiency (about 15%) was reached at low pH values (6.5–6.8 vs. 7.1), which was suitable due to the acidic pH of the mucus [43]. Finally, the mPEG coating of nanocomplexes allowed the protection of the cargo from nuclease digestion and from the process of nebulisation, so that well-functioning nanocomplexes were efficiently aerosolised. The same group [44] generated a PEGylated Cs/CRISPR/Cas9 complex by thin film freeze-drying to be used in dry powder inhalers, which showed higher physical and chemical stability than nebulisers [45]. The transfection efficiency was assessed in HEK-293 cells and was found to depend on the type and concentration of cryoprotectants such as sugars [44]. Overall, results in these studies suggested that PEGylated CsNPs might represent an optimal gene editing system through the mucus. However, due to its increased hydrophilicity, this carrier may be used for other biomedical applications such as its delivery via intravenous administration.

#### 3.3.3. Chitosan-Coated Red Fluorescent Protein

Chitosan-coated red fluorescent protein (RFP) (Figure 7C) was used by Quiao et al. [46] as a carrier of mature Cas9 RNPs and single strand DNA (ssDNA). To increase the electrostatic interaction with chitosan, the Cas9 enzyme was modified by the insertion of twenty glutamate residues (which increased the negative charges) and three repeating nuclear localisation signals (NLSs), which conferred nuclear targeting along with further negative charges [47,48]. Since RFP is negatively charged, it was encapsulated in a shell of protonated low MW chitosan (about 1 kDa), resulting in NPs with a size lower than 10 nm and a highly positive distribution of charges (+50 mV). Encapsulation of the RNPs and ssDNA led to an increase in the size of NPs (up to 200 nm), while becoming negatively charged (about −10 mV). Following in vitro cellular uptake experiments, it was observed that the nanocomplexes were localised around the nucleus of cells, and their uptake was due to a caveolae- and micropinocytosis-mediated endocytosis.

### 3.4. Strategies Combining Higher Specificity and Stability

#### 3.4.1. Trimethyl Chitosan with Stimulus-Triggering and Targeting Responses

TMC is a derivative of chitosan characterised by permanent positive charges conferred by the presence of methyl groups conjugated to the amino groups of chitosan. Advantages of the permanent positive charges include improved solubility in a broad range of pH values, absorption efficiency, and mucoadhesion [49]. The hydrophilic properties of TMC were employed by Li et al. [50] to increase the condensation efficiency and protection from nucleases of nucleic acids. Moreover, given the advantage of TMC to be further modified with functional groups [51], 2-(diisopropylamino) ethyl methacrylate (DPA) and folic acid (Fa) were incorporated into the TMC structure (Figure 8A). DPA and Fa conferred stimulus-triggered drug release [52] and targeting properties, respectively, given the overexpression of Fa receptor in many tumour cells [53]. This novel system was used for the co-delivery of doxorubicin (Dox) and Survivin CRISPR/Cas9-expressing plasmid deoxyribonucleic acid (sgSurvivin pDNA), aiming to down-regulate the expression of Survivin (involved in the inhibition of apoptosis of tumour cells) and enhance the effect of the chemotherapeutic agent, Dox [50]. It was shown that the treatment of 4 T1 cells with the loaded NCs achieved a high nuclear delivery efficiency of the cargo reaching values of about 80% after 8 h incubation. Similar results were also obtained in vivo, following intravenous administration to mice bearing breast tumours where the expression of Survivin in the tumour tissue was found to be significantly lower than the control (37.1% vs. 68.4%). In addition, significant tumour inhibition was recorded compared to the group treated with free Dox and NPs, where the Fa ligand was absent. The advantage of the Fa ligand was even more accentuated following genome editing efficiency evaluation [50].

#### 3.4.2. Carboxymethyl Chitosan with Cellular and Nuclear Targeting

Similar to TMC, carboxymethyl chitosan (CmCs) is a widely used derivative of chitosan due to its improved solubility in a wide range of pH values. Indeed, carboxyl groups have a pK_a_ of approximately 4.5, meaning that they become deprotonated, and hence negatively charged, at physiological pH, increasing the dissolution of CmCs. This effect is also given by the amino groups of chitosan that become protonated in the acidic tumour environment [54]. For this reason, CmCs is often employed in gene and drug delivery systems that require a pH-dependent response [13]. Moreover, carboxyl groups of CmCs can be further modified with other molecules or polymers to enhance the efficiency of the gene delivery system [55]. Regarding the delivery of the CRISPR/Cas9 complex, Liu et al. [56] generated a dual-targeting delivery system where the shell was composed of CmCs further functionalised with two molecules: biotin, a widely used target ligand of several tumour cells [57] and AS1411 (Figure 8B). The core of the vector was made of protamine sulphate, calcium carbonate, and calcium phosphate that co-precipitated with the CRISPR/Cas9 complex. In this study, the complex was directed against a cyclin-dependant kinase (CDK) overexpressed and overactive in tumour cells [58]. The results showed that the delivery of the plasmid to the nucleus of tumour cells led to a 90% reduction in the expression of the CDK and other proteins related to tumour development, such as Survivin. The same authors [59] developed another vector wherein CmCs was functionalised with AS1411 and a cell-penetrating peptide, transactivating transcriptional activator (TAT) to improve cellular uptake and endosome escape (Figure 8C) [60]. The CRISPR/Cas9 complex used was against the gene encoding β-catenin, a molecule involved in the proliferation, migration, and invasion of tumour cells. A successful knockout of the protein was observed, suggesting the ability of the vector to mediate genome editing in the nuclei of cancer cells.

### 3.5. Strategy Boosting the Buffering Ability

Calcium phosphate (CaP) was shown to be a cost-effective and efficient gene delivery system in vitro, but it has limitations such as easy degradation of the genetic material upon in vivo application [61]. Therefore, Rabiee et al. [62] investigated the synergetic effect of CaP and chitosan to condensate pCRISPR in a form of nanocomplexes. The main advantage of using both chitosan and CaP relied on the boosted buffering ability that causes endosome bursting following the protonation of both molecules. As expected, the presence of chitosan increased the zeta potential of the NPs, resulting in increased genetic material bonded. The GFP expression was evaluated in HEK-293 cells and reached values of 25% in the presence of chitosan. The authors suggested that the presence of calcium induced a charge separation on chitosan surface which ultimately led to reduced size of the nanocomplexes, improving their transfection efficiency [62].

### 3.6. Strategy Combining Higher Specificity, Stability, and Buffering Ability

The buffering ability of the Cs-based NC was boosted by Liu et al. [63] through the incorporation of KALA, a positively charged, cell-penetrating, and endosomolytic peptide [64]. Furthermore, the negatively charged aptamer AS1411 was also employed to increase the specificity of the CRISPR/Cas9 plasmid nanocarriers based on CmCs (Figure 9). In this study, the plasmid was directed to knockout CDK11, overexpressed in cancer cells where it induces cell growth [65]. Cellular uptake results showed that, as expected, the presence of both ligands on the surface of NPs optimised the delivery efficiency of CDK11 loaded NPs in the nuclei of cancer cells compared to normal cells. Indeed, upon genome editing, the expression of CDK11 was dramatically decreased in vitro (>75%) followed by restored tumour-suppression protein p53 and immune-related proteins, suggesting that these multifunctional NPs based on Cs were able to reverse tumour-induced immunosuppression and prevent tumour development in vitro [63].

## 4. Discussions

CRISPR/Cas9 technology is a cutting-edge gene editing approach due to its simple and cheap mechanism to edit genomes with high precision [7]. In 2016, it was approved for clinical trials [66] and since then it has attracted the worldwide interest for its application in the treatment of many types of cancer and other diseases with a genetic origin, such as cystic fibrosis. However, the most challenging factor for its employment is the delivery of the system to the cells. Indeed, after intravenous administration, the CRISPR/Cas9 system needs to have a prolonged lifetime in the blood, hence it needs to be protected by extracellular nucleases that degrade it. Moreover, the high MW and the strong negative charges of the phosphate backbone in the sgRNA hinder its passage through the cellular membrane. Once this barrier is overcome, the complex needs to reach the nucleus where it can integrate itself with the genome to enable gene editing. However, before achieving this, it also needs to survive the endocytic pathway that otherwise leads to its degradation. To overcome all these barriers, the development of an efficient CRISPR/Cas9 delivery system is needed. In particular, to be used in clinical trials, both its safety profile and the efficient delivery of the cargo must play key roles in circumventing insertional mutagenesis, immunogenicity, or off-target effects. Within all the possible options for the development of such a vector [5], in the last few years, chitosan-based CRISPR/Cas9 carriers have attracted the interest of the scientific community due to the innumerable advantages that this polymer can offer. This review summarises the current evidence about the application of chitosan-based CRISPR/Cas9 delivery systems (Table 1).

Three studies have shown that chitosan used on its own was able to achieve high transfection efficiency of CRISPR/Cas9 to the target site, implying high stability of the vector [23,25,27]. However, these investigations were performed in vitro using only the cell lines of interest, hence limited information is available regarding the targeting ability of the vector in vivo. Moreover, further studies on the stability of the CRISPR/Cas9 vector in the physiological fluids are needed. Indeed, it is key to consider that the positively charged surface of CsNPs attracts negatively charged proteins, such as albumin, the most abundant protein found in the blood circulation [67,68]. This leads to the formation of a “protein corona” which limits CsNP’s in vivo applications [37]. For instance, Alallam et al. [27] performed this investigation on pristine CsNPs and suggested that the formation of protein corona on CsNPs was also responsible for a reduced in vitro cellular uptake. Therefore, to overcome this effect, the functionalisation of chitosan structure is needed [37]. Indeed, in the context of CRISPR/Cas9 delivery, chitosan-based vectors have been modified to achieve high stability and tumour specificity, for instance by simultaneously conjugating chitosan to trimethyl groups and the Fa ligand [50]. Some of these types of modifications of chitosan structure were also used in the context of vaccines [69]. In particular, since CsNPs show excellent ability to improve mucosa absorption and stability, they have been used as a DNA vaccine carrier. However, the oral delivery of DNA-loaded CsNPs is hampered by the DNA degradation in the gut, hence CsNPs need to be coated with materials that protect the encapsulated DNA. For instance, alginate has been used to coat CsNPs since it is stable at very acidic pH such that of the stomach [70].

In the context of CRISPR/Cas9 delivery, the use of chitosan needs to overtake the advantages of other materials also proposed as non-viral CRISPR/Cas9 vectors. For instance, PEI is widely used in gene delivery since it is considered one of the main suitable polymers for transfection studies [71]. PEI’s chemical structure is very similar to that of chitosan: it is a strongly cationic polymer, due to the presence of several amine groups on its backbone and hence, it can strongly interact with negatively charged DNA forming stable polyplexes [72]. Moreover, due to the amino groups, PEI is characterised by a potent proton sponge property [73], which is even stronger than that of chitosan [13]. Indeed, Rabiee et al. [62] proposed to enhance this property by the condensation of chitosan with CaP for the delivery of CRISPR/Cas9. Moreover, PEI-based polyplexes are engulfed by cells via both clathrin- and caveolae-dependent pathways. The advantage of the latter is the non-linkage with lysosomes, so that the genetic material is protected by lysosomal enzymes [74]. However, the cellular uptake of CRISPR/Cas9 loaded chitosan-coated RFP was shown to occur through a caveolae and micropinocytosis mediated pathway [46], suggesting that also CsNPs may escape the lysosomes. In contrast to chitosan’s natural origin, PEI is a synthetic polymer, implying that during the process of manufacturing, the correct choice of the degree of branching and the MW is needed, to avoid an effect on the cell viability [72,75]. Lipofectamine^TM^ is a commercial nucleic acid vector, also widely used in the delivery of CRISPR/Cas9 since it employs cationic liposomes interacting strongly with the nucleic acids, hence protecting them from degradation [10]. Moreover, Lipofectamine^TM^ also circumvents the microtubule-mediated intracellular pathway, in turn avoiding lysosomal degradation, thereby enhancing the transfection efficiency [76]. However, in their study, Srivastav et al. [25] showed that treatment with CRISPR/Cas9 loaded chitosan-coated PLGA NPs induced a comparable transfection efficiency and stronger gene silencing of Lipofectamine 3000. Conversely, He et al. [59] proved that CmCs functionalised with AS1411 and TAT and loaded with CRISPR/Cas9 produced a higher cellular uptake toward tumour cells than non-tumour ones, when compared to Lipofectamine 3000, perhaps due to the presence of the tumour targeting ligand, AS1411. Moreover, an effect on cytotoxicity was recorded upon treatment with Lipofectamine 2000 and 3000, while better biocompatibility was observed upon treatment with chitosan-based CRISPR/Cas9 vectors [59].

As stressed by Alallam et al. [27] the physicochemical properties of Cs-based NPs play a key role in determining a high cellular uptake and, as consequence, a high transfection efficiency of CRISPR/Cas9. Table 2 summarises the most important physicochemical properties (size and zeta potential) of the various Cs-based CRISPR/Cas9 delivery systems explored so far.

The size of Cs-based NPs mostly employed for the delivery of the CRISPR/Cas9 system is in the range of 200 nm, since this is the threshold size for recognition by most of the cells [77]. However, the size might be higher when NPs are coated with a polymer, such as chitosan [25,27]. In this case, the positive charges on chitosan play a key role in the internalisation since determining NP-cell interactions. A zeta potential higher than +30 mV or lower than −30 mV is generally considered as a stable colloid [78]. However, concerns may arise about the cell toxicity, hence a suitable kinetic stability of NPs needs to be achieved without affecting the cell viability. For instance, the zeta potential obtained by Alallam et al. [27] made the NPs more stable while affording the protection of plasmid DNA when in the serum. However, the zeta potential decreased to +6 mV upon serum proteins’ incubation, these being negatively charged, while CsNPs were more susceptible to aggregation. Therefore, protein corona investigations need to be performed before bringing Cs-based NPs to in vivo studies, so as to predict their behaviour in physiological conditions, especially when chitosan is used as a coating material.

Most of the nonviral vectors employed to deliver CRISPR/Cas9 show good performance, but they can only be employed in vitro, ex vivo, or following local in vivo administration whilst the systemic administration remains unsatisfactory [2]. Regarding chitosan-based CRISPR/Cas9 vectors, most of the research investigated the transfection efficiency in vitro, especially toward the HEK-293 cell line. This cell line, along with the human cervical adenocarcinoma cell line (HeLa), represents a well-established cell line for gene transfection investigations [10]. However, the HEK-293 cell line is not a cancer cell line, hence the transfection efficiency might be lower than cancer cells, as shown by Khademi et al. [32] This suggests that further in vitro studies should be performed using cancer cells while using the HEK-293 cell lines as a control. Moreover, in vivo studies are needed to establish the successfulness of these chitosan-based vectors in gene delivery. Zhang et al. [29], Khademi et al. [32], and Li et al. [50] were the only three investigations that tested the novel chitosan-based CRISPR/Cas9 vectors for their in vivo antitumour and gene editing efficacy. Indeed, acting upon in vivo studies is key to have a high specificity of the vector, to reduce the off-target effects. NPs-loaded CRISPR/Cas9, based on Cs conjugated to β-galactose-carrying La, were tested on the hepatoma mice model, which showed inhibition of the tumour growth because of an efficient genome editing ability [29]. Khademi et al. [32] assessed the specificity ability of Cs CRISPR/Cas9 NPs decorated with HA and AS411 to suppress the tumour growth which was found to be up to 90%. Li et al. [50] employed CRISPR/Cas9 vectors based on TMC modified with Fa and loaded with Dox and sgSurvivin pDNA, which showed high biocompatibility in the blood and high targeting efficiency. Therefore, to move this application of chitosan toward clinical studies, the efficiency of the generated vector also needs to be investigated in vivo. So far, only these three chitosan-based systems are ready to be translated into clinical studies [29,32,50]. 

## 5. Conclusions

The most recent chitosan-based approaches for delivering CRISPR/Cas9 gene editing technology have been summarised and discussed in comparison with the commercial lipid-based CRISPR/Cas9 delivery systems including Lipofectamine^TM^ and other polymer-based vectors. Being a natural cationic polymer, chitosan confers several advantages to the nonviral vector, even though its pristine chemical structure shows some limitations such as buffering ability, poor specificity, and stability upon intravenous administration. However, due to the presence of functional groups on its structure, chitosan can be easily modified by the addition of ligands, or hydrophilic groups, to overcome its limitation and improve its properties. Only few of the novel chitosan-based systems for CRISPR/Cas9 delivery were tested in vivo. Results suggested that Cs conjugated to β-galactose-carrying La, TMC modified with Fa and DPA and Cs modified with Ha and AS1411 were suitable CRISPR/Cas9 vectors following intravenous administration. The remaining NP formulations have been tested only in vitro and some of them (including CmCs functionalised with AS1411, TAT and chitosan coated PLGA NPs) showed similar or better transfection efficiency than the commercially available Lipofectamine^TM^. Preclinical investigations are encouraged to move these formulations toward the clinical applications.

## Figures and Tables

**Figure 1 pharmaceutics-14-01840-f001:**
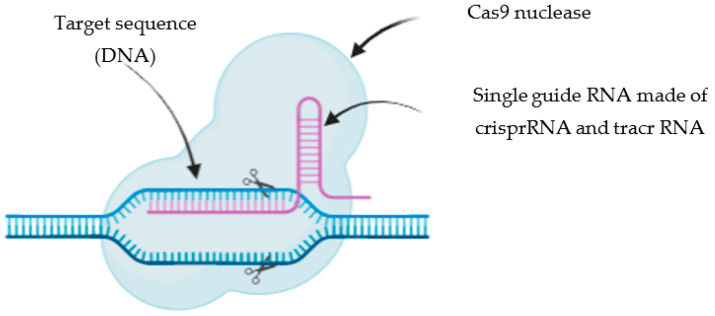
CRISPR/Cas9 mechanism of action. The specific sites on the target sequence (DNA) are recognised by the CRISPR of the single guide RNA. The Cas9 nuclease then cleaves the target sequence, introducing double strand breaks, which are then repaired by homology directed repair or non-homologous end joining.

**Figure 2 pharmaceutics-14-01840-f002:**
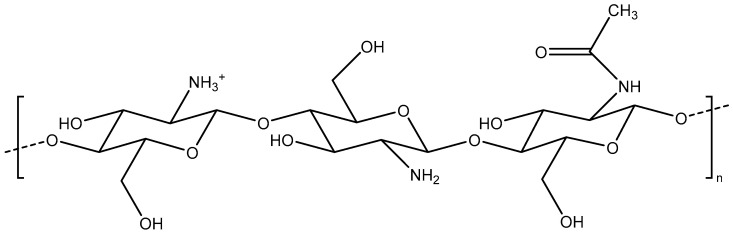
Chemical structure of chitosan.

**Figure 3 pharmaceutics-14-01840-f003:**
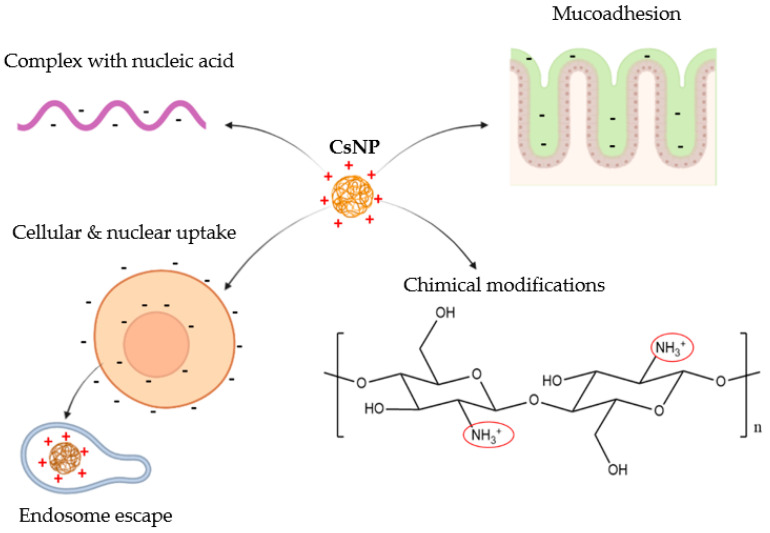
Chitosan nanoparticles’ (CsNP) properties conferred by the presence of positive charges of D-glucosamines: these allow CsNPs to complex with the negatively charged nucleic acid, increase mucoadhesion at the intestinal level, boost cellular and nuclear uptake following by endosome escape, and finally, the D-glucosamines allow chemical modifications to enhance chitosan chemical properties.

**Figure 4 pharmaceutics-14-01840-f004:**
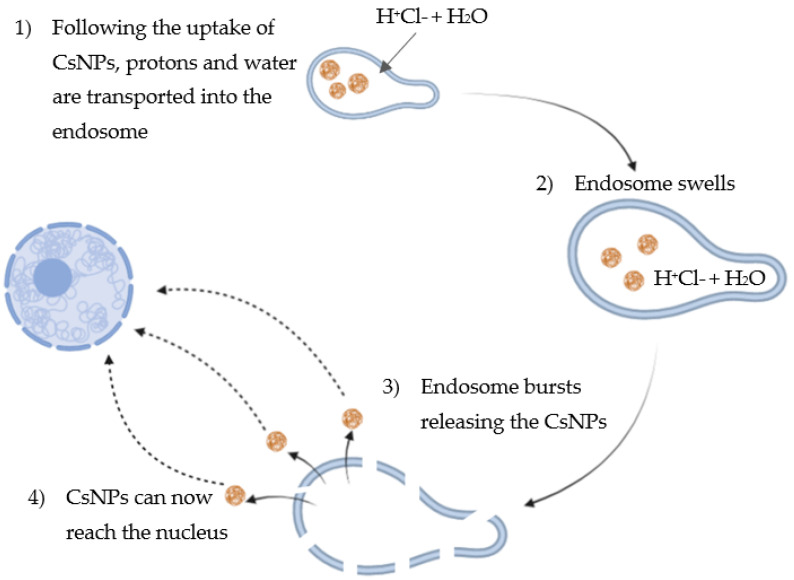
Graphical representation of the “proton sponge effect” following the uptake of CsNPs by the endosome: (1) protons are transported into the endosome along with chloride and water; (2) endosome swells; (3) endosome bursts releasing the CsNPs; and (4) CsNPs can now reach the nucleus.

**Figure 5 pharmaceutics-14-01840-f005:**
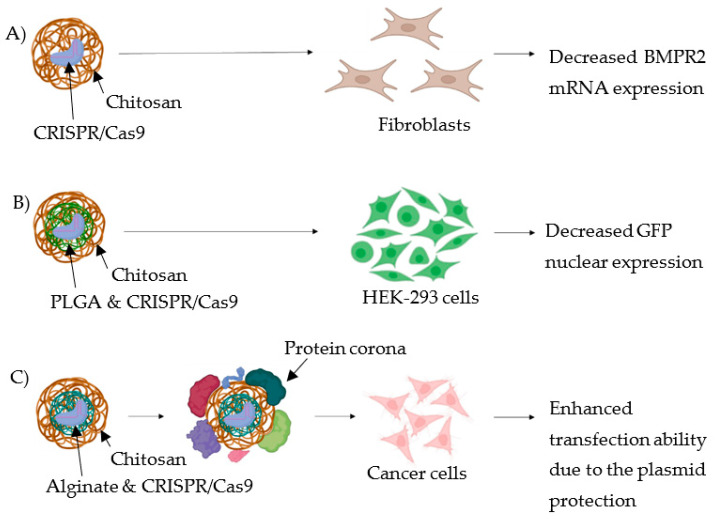
Strategies using pristine chitosan backbone. (**A**) Pristine CsNPs are used to transfect fibroblasts for the treatment of pulmonary arterial hypertension; results showed a decreased mRNA expression of the main gene, BMPR2, involved in the disease. (**B**) Chitosan is used as coating material of PLGA NPs complexed with CRISPR/Cas9; transfection of HEK-293 cells showed a decreased GFP nuclear expression. (**C**) Chitosan is used as coating material of alginate NPs complexed with CRISPR/Cas9; following formation of protein corona, treatment of cancer cells showed an enhanced transfection ability due to the plasmid protection from enzymatic degradation.

**Figure 6 pharmaceutics-14-01840-f006:**
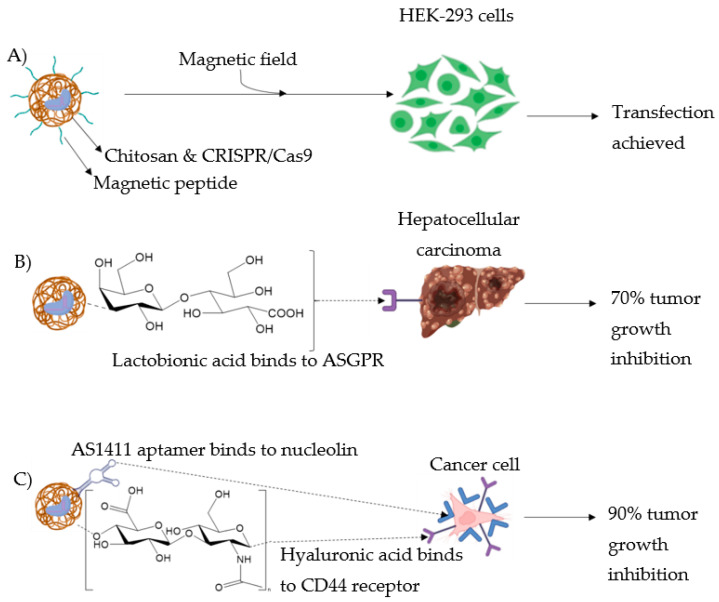
Strategies improving the specificity of CsNPs. (**A**) Upon application of a magnetic field, HEK-293 cells were successfully transfected using CRISPR/Cas9 complex immobilised on magnetic peptide-imprinted CsNPs. (**B**) Conjugation of lactobionic acid to chitosan-induced NPs to preferentially accumulate within ASGPR-overexpressing cells, so as to achieve up to 70% tumour growth inhibition. (**C**) Two ligands were integrated into CsNPs: AS411 aptamer and hyaluronic acid bind to nucleolin and CD44 receptors overexpressing cancer cells, respectively, so as to achieve up to 90% tumour growth inhibition.

**Figure 7 pharmaceutics-14-01840-f007:**
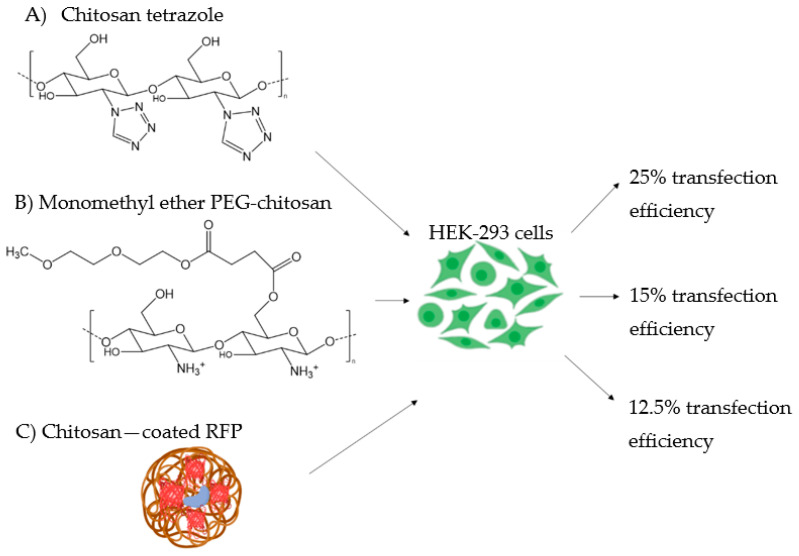
Strategies improving the stability of CsNPs. (**A**) Chitosan tetrazole; (**B**) monomethyl ether PEG-chitosan; and (**C**) chitosan-coated red fluorescent protein (RFP). The transfection efficiency of these strategies was tested using HEK-293 cells ranging between 12.5 and 25%.

**Figure 8 pharmaceutics-14-01840-f008:**
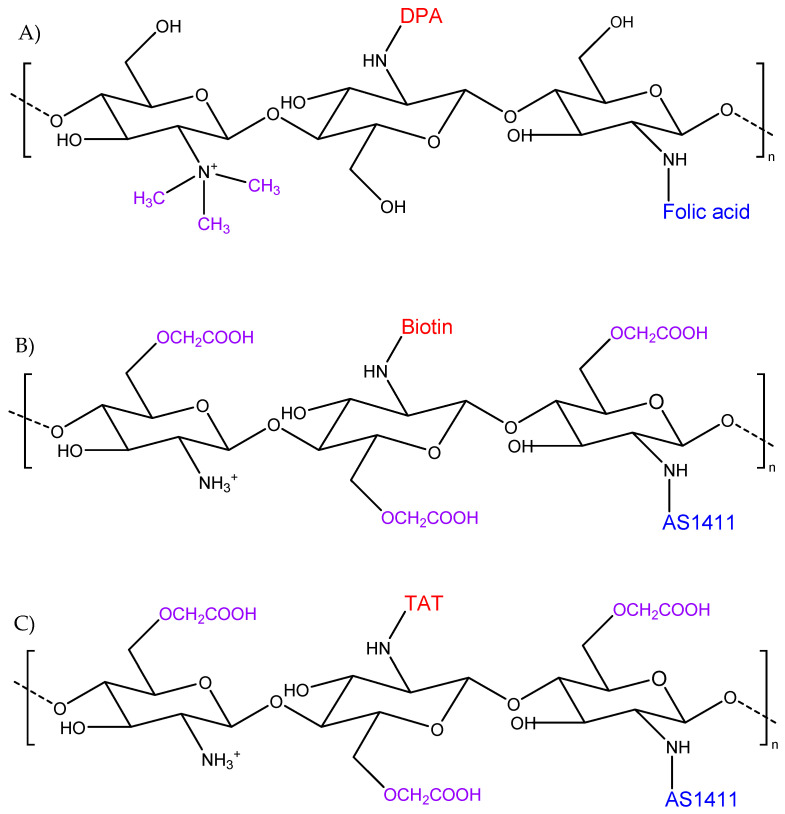
Strategies combining higher specificity and stability: (**A**) trimethyl chitosan modified with DPA and folic acid, (**B**) carboxymethyl chitosan modified with AS1411 aptamer and biotin, and (**C**) carboxymethyl chitosan modified with AS1411 aptamer and the transactivating transcriptional activator (TAT).

**Figure 9 pharmaceutics-14-01840-f009:**
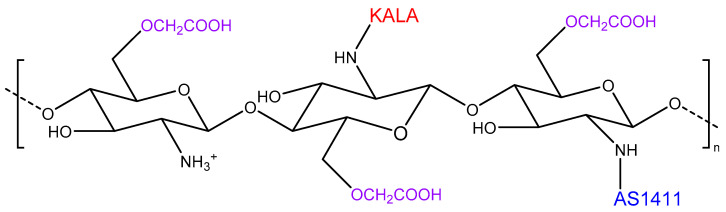
Carboxymethyl chitosan modified with AS1411 aptamer and the endosomolitic peptide, KALA, as a strategy combining higher specificity, stability, and buffering ability.

**Table 1 pharmaceutics-14-01840-t001:** Summary of the strategies adopted to increase the transfection efficiency of CRISPR/Cas9 complex by using chitosan-based vectors.

Properties of Chitosan	Strategies	Transfection Efficiency of CRISPR/Cas9	Reference
Pristine properties	CsNPs	Achieved in vitro	[23]
	Cs-coated PLGA NPs	>80% in vitro	[25]
	Cs-coated AG NPs	>20% in vitro	[27]
Improved specificity	Magnetic peptide-imprinted CsNPs	Achieved in vitro	[28]
	La-ligand conjugated CsNPs	39% in vitro 33% in vivo	[29]
	Ha and AS1411-decorated CsNPs	20–30% in vitro 90% in vivo	[32]
Increased stability	Cs tetrazole	>25% in vitro	[36]
	PEGylated Cs	~15% in vitro	[12,44]
	Cs-coated red fluorescent protein	~12.5% in vitro	[46]
Improved specificity and stability	DPA and Fa-ligand-modified TMC	~80% in vitro ~30% in vivo	[50]
	Biotin- and AS1411-modified CmCs	90% in vitro	[56]
	AS1411 and TAT-modified CmCs	~30% in vitro	[59]
Boosted buffering ability	Cs and calcium phosphate NPs	25% in vitro	[62]
Improved specificity, stability, and buffering ability	AS1411 and KALA-modified CmCs	>75% in vitro	[63]

**Table 2 pharmaceutics-14-01840-t002:** Physicochemical properties of the CRISPR/Cas9 vectors.

Strategies	Size (nm)	Zeta Potential (mV)	Reference
CsNPs	Not specified	Not specified	[23]
Cs-coated PLGA NPs	~300	+32	[25]
Cs-coated AG NPs	~400	+33	[27]
Magnetic-peptide-imprinted CsNPs	Not specified	Not specified	[28]
La-ligand-conjugated CsNPs	~200	+26	[29]
Ha and AS1411-decorated CsNPs	~200	+20	[32]
Cs tetrazole	~80	+40	[36]
PEGylated Cs	~200	+20	[12,44]
Cs-coated red fluorescent protein	~200	+50	[46]
DPA and Fa-ligand-modified TMC	~200	+20	[50]
Biotin and AS1411-modified CmCs	~300	−9	[56]
AS1411 and TA-modified CmCs	~300	−7	[59]
Cs and calcium phosphate NPs	~150	+27	[62]
AS1411 and KALA-modified CmCs	~240	+17	[63]

## Data Availability

Not applicable.

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
