# Peer review of "Advances in Chitosan-Based CRISPR/Cas9 Delivery Systems"

_pharmaceutics, 2022, doi:10.3390/pharmaceutics14091840_

Round 1

Reviewer 1 Report (New Reviewer)

General editorial comments

This manuscript is well written and presented. The topic is very interesting and appropriate for

the journal, balancing overview with some details. The detailed descriptions of the different

types and chemistry of the developed systems and applications are instructive and accurate, and

the selection of references provides an unbiased overview of the field, highlighting different

research and scientific directions.

The figures, however, do not yet match the quality of the text. Figures need to be

improved and supported with more educational elements to ensure the overall highly instructive

character of this review article. There are some additional issues and questions (itemized below),

which I encourage the authors to address to improve the quality of this manuscript.

Strengths

• Clear and explicit written content

• Concise structure, in the general organization of the manuscript - and also, per

(sub)section

• Citation of relevant publications in the field

General weaknesses

• The structure of the review article in its current form is confusing – there are two

subsections with almost the same title, regarding the stability and specificity, a separate

section for discussions and repetitive text.

• A limited number of figures that support the include rich information and text through the

manuscript. It would be very useful if the authors provided additional figures that cover

all the major sections and subsections included in the manuscript.

• It will be interesting to see a tabled comparison between the different studies that used

chitosan-based material for the delivery of CRISPR systems with a focus on how the

nanoparticulate features of chitosan provided the mean for enhanced delivery and activity

of CRISPR.

• How much these nanosystems were successful in the delivery of other DNA drugs (e.g.,

DNA vaccines, plasmids, etc.)? How and why will they be better for CRISPR delivery?

Readers will be interested to see such a high-level analytical point of view.

• The text needs an additional language check for grammatical errors and typos.

• The Figures: Almost all labels in the figures are too small to read (even for people with

perfect vision). Many of the graphics in the figures are too small as well or have poor

contrast and are complex (too many details to follow in the same figure).

Specific weaknesses

• None was noted.

Author Response

Dear Reviewers,

Thanks for your suggestions and for your prompt replies.

In response to your comments the following actions were taken.

Reviewer #1:

-It would be very useful if the authors provided additional figures that cover all the major sections and subsections included in the manuscript.

Additional figures have been added.

-The structure of the review article in its current form is confusing – there are two subsections with almost the same title, regarding the stability and specificity, a separate section for discussions and repetitive text.

The subsections have been renamed.

- It will be interesting to see a tabled comparison between the different studies that used chitosan-based material for the delivery of CRISPR systems with a focus on how the nanoparticulate features of chitosan provided the mean for enhanced delivery and activity of CRISPR.

Table 2 has been added in the discussion section.

- How much these nanosystems were successful in the delivery of other DNA drugs (e.g., DNA vaccines, plasmids, etc.)? How and why will they be better for CRISPR delivery? Readers will be interested to see such a high-level analytical point of view. 

This has been addressed in the discussion section.
- The text needs an additional language check for grammatical errors and typos.

The language has been checked.

-The Figures: Almost all labels in the figures are too small to read (even for people with perfect vision). Many of the graphics in the figures are too small as well or have poor contrast and are complex (too many details to follow in the same figure).

All figures have been made again with higher contrast and less complexity.

Reviewer 2 Report (Previous Reviewer 1)

I believe that the review is interesting and well organized and therefore can be accepted in this form

Author Response

Dear Reviewer,

Thanks for your suggestions and for your prompt replies.

Reviewer 3 Report (New Reviewer)

This review presents specific examples and explanations of CRISPR/Cas9 gene editing tool using the various chitosan backbone system. In particular, it explains in detail the advantages of delivering gene editing materials such as plasmid by external stimuli. This is expected to be of great help to related researchers.

However, it would be better if some explanations were added.

1.     Recently, many problems have been reported in gene editing through plasmid gene delivery. For example, there are problems with integration, and high off-target effects. Therefore, many studies are underway to overcome this by delivering the Cas9/sgRNA structure in the form of a protein rather than a plasmid. The author, nevertheless, should write the advantages of gene editing through plasmid delivery in more detail in the introduction part.

2.     In the comment on No. 1, there are a lot of recent studies on protein-based Cas9/sgRNA delivery, and it is desirable to add an example of delivery using a Chitosan backbone. (Jie Qiao et. al., Chem. Comm. 2019, etc.)

Author Response

Dear Reviewer,

Thanks for your suggestions and for your prompt replies.

In response to your comments the following actions were taken.

Reviewer #3:

- Recently, many problems have been reported in gene editing through plasmid gene delivery. For example, there are problems with integration, and high off-target effects. Therefore, many studies are underway to overcome this by delivering the Cas9/sgRNA structure in the form of a protein rather than a plasmid. The author, nevertheless, should write the advantages of gene editing through plasmid delivery in more detail in the introduction part.

More details have been added in lines 46-56.

- In the comment on No. 1, there are a lot of recent studies on protein-based Cas9/sgRNA delivery, and it is desirable to add an example of delivery using a Chitosan backbone. (Jie Qiao et. al., Chem. Comm. 2019, etc.)

A section 3.1 was added with the subtitle: Strategies using pristine chitosan backbone

Reviewer 4 Report (New Reviewer)

Caprifico et al. have reviewed the use of chitosan-based carriers for the delivering of a CRISPR/CAS9 complex, which is a promising strategy to treat genetic diseases. Overall, the review is well written and comprises an introduction that provides a good view of the principles and potential of this technique, followed by other sections where varied strategies employing chitosan and its nanoparticles have been properly described. In the discussion section, the authors have highlighted the advantages of chitosan compared to other vectors and the positive results regarding the in vitro transfection efficiencies. The review discusses the limitations and points out the still incipient number of in vivo studies, which are pivotal to test the effectiveness of chitosan-based nanoparticles for the targeted delivery of CRISPR/CAS9 complexes.

Although the number of studies using chitosan nanoparticles for the CRISPR/CAS9 delivery is limited, in my opinion the most important studies have been covered and the authors offered a good overview of the potential of chitosan for this purpose. I would recommend that the authors insert the studies listed below in Section 3 and Table I. Based on the comments above, I would recommend the publication of this article after minor revision.

1)       Carbohydrate Polymers :10.1016/j.carbpol.2022.119691

2)       JOURNAL OF DRUG DELIVERY SCIENCE AND TECHNOLOGY: 10.1016/j.jddst.2021.102910

3)        Journal of Controlled Release : 10.1016/j.jconrel.2018.10.018

 Line 210: What is the  meaning for La ?

Author Response

Dear Reviewer,

Thanks for your suggestions and for your prompt replies.

In response to your comments the following actions were taken.

Reviewer #4:

- I would recommend that the authors insert the studies listed below in Section 3 and Table I.

1) Carbohydrate Polymers :10.1016/j.carbpol.2022.119691

2) JOURNAL OF DRUG DELIVERY SCIENCE AND TECHNOLOGY: 10.1016/j.jddst.2021.102910

3) Journal of Controlled Release: 10.1016/j.jconrel.2018.10.018

These studies have been inserted inside the text.

- Line 210: What is the meaning for La?

The meaning of La was added.

Round 2

Reviewer 1 Report (New Reviewer)

Nothing at this point.

This manuscript is a resubmission of an earlier submission. The following is a list of the peer review reports and author responses from that submission.

Round 1

Reviewer 1 Report

The manuscript summarizes the latest studies on chitosan-based nanoparticles and chitosan derivatives for the delivery of the CRISP / Cas9 gene. The manuscript is well organized, perhaps it would be better to include more figures describing the cited studies. However, I believe that the manuscript can be accepted for publication.